# Probing warm dense matter using femtosecond X-ray absorption spectroscopy with a laser-produced betatron source

B. Mahieu[1], N. Jourdain[2,3], K. Ta Phuoc[1], F. Dorchies[2], J.-P. Goddet[1], A. Lifschitz[1], P. Renaudin [3] & L. Lecherbourg[3]

Exploring and understanding ultrafast processes at the atomic level is a scientific challenge. Femtosecond X-ray absorption spectroscopy (XAS) arises as an essential experimental probing method, as it can simultaneously reveal both electronic and atomic structures, and thus potentially unravel their nonequilibrium dynamic interplay which is at the origin of most of the ultrafast mechanisms. However, despite considerable efforts, there is still no femtosecond X-ray source suitable for routine experiments. Here we show that betatron radiation from relativistic laser−plasma interaction combines ideal features for femtosecond XAS. It has been used to investigate the nonequilibrium dynamics of a copper sample brought at extreme conditions of temperature and pressure by a femtosecond laser pulse. We measured a rise-time of the electron temperature below 100 fs. This experiment demonstrates the great potential of the table-top betatron source which makes possible the investigation of unexplored ultrafast processes in manifold fields of research.

[1] LOA, ENSTA ParisTech, CNRS, Ecole Polytechnique, Université Paris-Saclay, 828 Boulevard des Maréchaux, 91120 Palaiseau, France. [2] Université de Bordeaux, CNRS, CEA, CELIA (Centre Lasers Intenses et Applications), UMR 5107, 33400 Talence, France. [3] CEA-DAM-DIF, 91297 Arpajon, France. These authors contributed equally: B. Mahieu, N. Jourdain.  Correspondence and requests for materials should be addressed to B.M. (email: benoit.mahieu@ensta.fr)

X-ray absorption spectroscopy (XAS) techniques are essential tools for probing both electronic and atomic structural properties of matter. Including X-ray absorption near-edge structure (XANES) and extended X-ray absorption fine structure, they are exploited in a wide range of applications in coordination chemistry, gas-phase systems, materials science or for the study of complex biological samples[1]. XAS has been extensively developed at synchrotrons, taking advantage of the broadband photon spectrum lying in the keV region, where most of elements absorption edges are located. When one aims at studying the temporal response of a sample after ultrafast excitation, referring to the time-resolved XAS technique, the temporal resolution of the observed phenomena becomes limited by the duration of the synchrotron pulses, that is ~10–100 ps, or the temporal resolution of a streak camera, generally few ps. That leaves veiled the very first moments where the out-of-equilibrium interplay between still frozen atoms and strongly modified electrons drives the ultrafast processes. To date, two main solutions were tested to circumvent this limitation: synchrotron slicing, where a few hundreds femtosecond temporal slice of the pulse is extracted to probe the sample at the cost of a greatly reduced photon flux[2,3], and wavelength scanning on a X-ray free-electron laser (XFEL) that requires a large number of shots to recover a single absorption spectrum[4]—XFEL features being more appropriate for ultrafast X-ray diffraction[5,6] due to a narrow spectrum. Both options rely on the use of expensive large-scale research instruments, providing limited access. Furthermore, the necessary number of shots mostly limits the range of systems and regimes that can be studied to continuously renewed targets, for example in liquid jets, or reversible phase transitions. On XFEL, investigation of a nonreversible process was albeit reported by means of dispersive measurements, that remain in any case limited to a few eV bandwidth and must circumvent the inherent shot-to-shot spectral fluctuations of such a source[7]. Finally, thermal plasma laser-based X-ray sources can provide broad spectra but with a duration down-limited to the picosecond[8] while most advanced high-harmonic generation sources—driven by few-cycle, near-infrared laser systems—demonstrated their capabilities at photon energies up to few hundreds of eV[9].

Here we present the demonstration of femtosecond-resolved XAS using a table-top X-ray source naturally combining broad spectrum and femtosecond duration with required features in terms of stability, photon flux, and inherent pump-probe synchronization. Our scheme relies on the betatron radiation from laser−plasma acceleration[10]. While this source was to date too unstable for carrying out any realistic XAS study, we recently demonstrated the production of stable betatron radiation[11] making it now possible[12]. Time-resolved XAS experiment has been performed to investigate at the atomic scale the ultrafast dynamics of a copper foil brought from solid to warm dense matter (WDM) by a femtosecond optical laser pulse. Such study has been selected as a paragon of a strongly excited system, leading to the full ablation of the heated sample after a single shot.

## Results

### Description of the application

WDM is a subject of increased interest due to its importance for planetary physics[13,14], inertial confinement fusion research[15], and material science[16,17]. WDM lies between solid and plasma, with a density close to the solid one and a temperature of a few $10^4$ K. It is characterized by a partial disorder but with strong atom correlation and electron degeneracy, which makes it challenging to simulate and predict properties. When produced in the laboratory by femtosecond isochoric heating of a solid foil, the energy is suddenly deposited in the electrons and is homogenized along the thickness in a femtosecond time scale[18,19]. That leads to strong out-of-equilibrium situations where the electron temperature can reach tens of thousands K while the lattice is still cold. The electron−ion thermal equilibration follows on a longer time scale (a few ps). The electron dynamics of WDM has been experimentally studied from XAS, but so far with only a few picosecond resolution, thus limiting the investigation to the long-lived relaxation dynamics of electron temperature[20]. Direct experimental investigation of femtosecond dynamics of electron heating and thermalization was not accessible so far, so that the determination of the maximum temperature achieved has remained unreachable until now.

### Molecular dynamics simulations

Figure 1 shows results of ab initio molecular dynamics simulations of the system under study, corresponding to three successive snapshots: cold lattice before heating (Fig. 1a), strong out-of-equilibrium at the femtosecond time scale (Fig. 1b), and electron−ion thermal equilibration in the picosecond regime (Fig. 1c). The copper $L_3$ and $L_2$ edges observed around 932 and 952 eV (Fig. 1d) respectively correspond to transitions from the $2p_{3/2}$ and $2p_{1/2}$ core levels up to unoccupied electron states in the continuum, just above the Fermi energy $E_F$. When the copper sample is heated by the pump pulse, electrons below $E_F$ start to be excited up to higher energy levels, leaving some unoccupied states in the $3d$ band whose upper part is located ~1–2 eV below $E_F$. These unoccupied states allow new transitions from initial $2p$ to final $3d$ states, leading to additional absorption observed a few eV below the L-edges[21] (Fig. 1e). This structure is called pre-edge. It characterizes the ultrafast electron excitation and vanishes at picosecond time scales (Fig. 1f).

### XAS measurements

Figure 2 presents the experimental setup we implemented for carrying out time-resolved XAS with the betatron source. Measurements of absorption spectra of the copper sample are shown in Fig. 3a: three selected pump-probe delays are illustrated, together with a "cold" absorption spectrum (without pump pulse).

For negative delays, the probe pulse arrives before the pump pulse and the spectrum remains unchanged from the cold situation. For positive delays, an increasing absorption is observed below the cold absorption edges: this is the pre-edge structure predicted by the theory. Figure 3b shows the difference between the measured absorption spectra of heated sample and the cold spectrum. It clearly emphasizes the presence of the rising pre-edge during the first hundreds fs that follow the pump laser irradiation.

### Electron temperature evolution

It has been recently shown that one can retrieve the electron temperature $T_e$ through this pre-edge[22]. Taking advantage of careful molecular dynamics simulations[23], we have thus deduced the time evolution of $T_e$ from the pre-edge spectral integration evaluated for each spectrum. The results are shown in Fig. 4. The electron temperature evolution is characterized by a fast femtosecond increase (corresponding to the sample heating) followed by a longer picosecond decrease (electron−ion thermal equilibration). Looking carefully at the first time steps, the rise-time is estimated at $\tau_{rise} = 75 \pm 25$ fs RMS, a little larger than the 30 fs expected from the heating pulse duration. In the same figure, we report the time evolutions of $T_e$ and $T_i$ calculated from a two-temperature hydrodynamic code (see Methods). The overall data are well reproduced by the model, including the maximal value of $T_e = 10,000 \pm 2000$ K. At the considered fluence, the calculation predicts that the melting occurs at ~1 ps after heating.

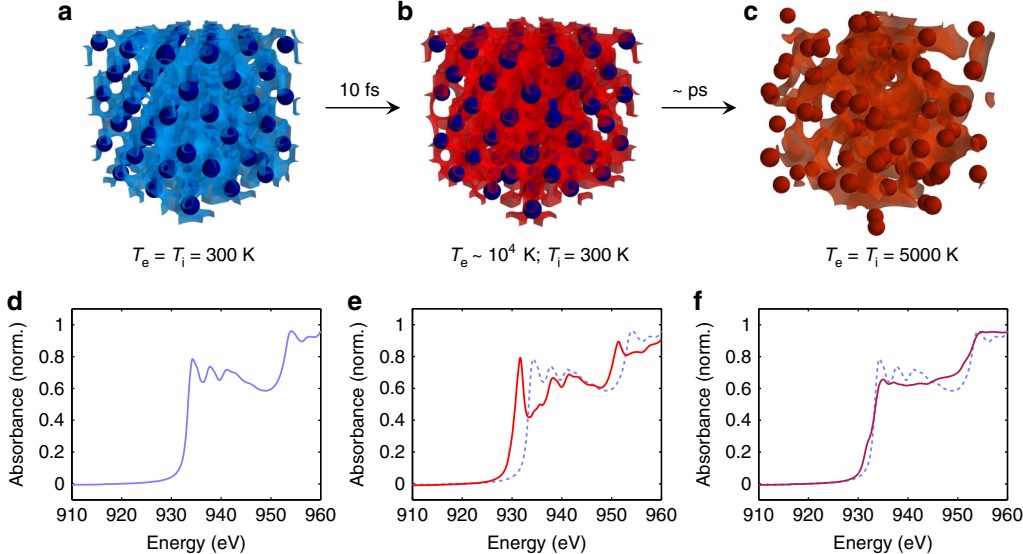

**Fig. 1** Numerical simulation of the ultrafast nonequilibrium transition of copper from solid to WDM. The energy of a femtosecond optical laser pulse is suddenly deposited in the electrons of the system (femtosecond scale), then progressively transferred to the lattice (picosecond scale). **a** Cold solid lattice before heating: the electron temperature $T_e$ equals the ion one $T_i$. **b** Just after heating, a strongly out-of-equilibrium situation is transiently produced where electrons are hot while the lattice is still cold and solid-like. **c** A few picoseconds after, the lattice disappears as electrons and ions progress up to their thermal equilibrium. **d–f** Calculated absorption spectra in the XANES region corresponding respectively to **a–c**. The cold XANES signal shown in **d** is reported in dashed line in **e**, **f**. See the Methods section for simulation details

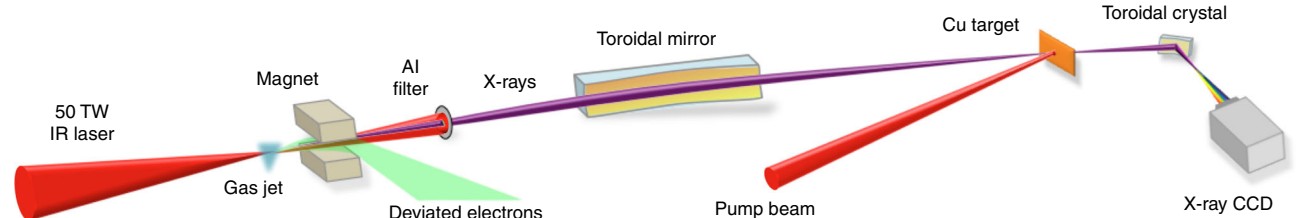

**Fig. 2** Setup of the experiment. A 50 TW, 30 fs laser pulse is focused onto a supersonic jet of 99% helium/1% nitrogen gas mixture. The interaction of the laser with the underdense created plasma yields the generation of a betatron X-ray pulse (see Methods for details). The latter is focused by a toroidal mirror on the Cu sample placed at normal incidence. A spectrometer composed of a toroidal crystal and an X-ray charge-coupled device (CCD) camera then record the transmitted spectrum. In parallel, a synchronized laser pulse (pump), with adjustable delay with respect to the X-ray pulse and with adjustable fluence, is used to heat the Cu sample up to the WDM regime. The absence of jitter is ensured by the fact that the pump laser and the laser-generated X-ray pulse originate from the same laser source

## Discussion

The overall temporal resolution of the measurement is the contribution of several parameters, among which the pump and probe durations—respectively measured at 30 fs and estimated at 9 fs FWHM (full-width at half maximum); see Methods. The pump beam forms a $\theta_x = 2.5°$ angle with the X-ray beam and the measured horizontal X-ray beam size is $\sigma_x = 200 \pm 60$ µm FWHM at the sample surface. This sets an additional geometrical contribution to the temporal resolution equal to $\sigma_x \cdot \theta_x / c = 30 \pm 10$ fs, but not yet enough to reproduce the observation. One must thus consider the physical process of thermalization occurring within the sample in order to understand the nature of $\tau_{rise}$. The energy deposition in the whole sample thickness can be described in two steps[24]. Electrons are first excited in the near-surface region within the penetration depth (~15 nm for copper at the wavelength of the pump laser); then, the energy is transported through the ballistic motion of electrons. Taking a characteristic electron velocity $v_e = v_F + \sqrt{k_B T_e / m_e}$, where $k_B$ is the Boltzmann constant, $m_e$ the electron mass and $v_F$ the Fermi velocity, leads to $v_e \simeq 10^6$ m s$^{-1}$. The critical parameter for the relevance of ballistic transport of the electrons is the electron mean free path $d$, which is governed by elastic, quasi-elastic, and inelastic scattering.

In copper $d = 70$ nm[24] and is approximately equal to the sample thickness so that homogeneous heating throughout the sample is ensured within a time scale comparable to the inelastic lifetime at optical excitation energies ($d/v_e \simeq 80 \pm 35$ fs). Electron heating is thus the main contribution to our observation. Furthermore, the expected value of $\tau_{rise}$ given statistically by the quadratic sum of the different contributions listed hereabove (pump duration, probe duration, temporal limit given by the pump-probe angle, electron heating time) is equal to $90 \pm 40$ fs, which is indeed in the range of our measurement.

In conclusion, we investigated the ultrafast and out-of-equilibrium transition of a copper foil brought from solid to WDM by a femtosecond laser pulse. X-ray absorption spectra are registered near the L-edge with unprecedented femtosecond temporal resolution. For this, we rely on the production of stable betatron X-ray pulses with few-femtosecond duration from a laser−plasma accelerator. We measured a rise-time of the electron temperature of $75 \pm 25$ fs, larger than the estimated temporal resolution. It is understood as the macroscopic energy diffusion time throughout the sample. Data are quite well reproduced by a two-temperature hydrodynamic calculation. It shows that the femtosecond resolution achieved allows to capture

out-of-equilibrium situations before the melting, thus providing access to the intimate understanding of ultrafast phase transition physics. This demonstration experiment opens the paths for studying matter under extreme conditions and ultrafast science in general[25]. In addition to its ultimate temporal resolution, betatron

is a table-top synchrotron-like X-ray source that now offers unique possibilities for a wide scientific community.

## Methods

**Laser beams.** The experiment was conducted at Laboratoire d'Optique Appliquée on a Ti:Sapphire laser system delivering $2 \times 1.5$ J pulses at a central wavelength of 810 nm. One arm is used for betatron X-ray generation. It was focused into a 3 mm supersonic 99%He−1%$N_2$ gas jet with a 1-m-focal-length off-axis parabola, to a focal spot size of ~15 μm (FWHM). A deformable mirror is used for improving the laser beam quality at focus. The second arm is used for sample heating. The laser presents a super-Gaussian profile with 48 mm $1/e^2$ diameter. A 10-mm iris is placed 7 m before the sample, forming a top-hat intensity distribution. The iris plane is imaged onto the sample by means of a near-normal incidence 25-mm-focal-length spherical mirror placed 26.8 mm in front of the sample. This allows a transversely homogeneous heating of the sample over a $\pi \cdot 400$ μm$^2$ sample surface. The pump beam fluence on the sample was controlled by finely tuning the pump beam energy. The maximum pump fluence was 6.5 J cm$^{-2}$. The nominal laser pulse duration was measured to be $30 \pm 5$ fs (FWHM) by a spectral-phase interferometry for direct electric field reconstruction (SPIDER) apparatus. Grating spacing of the laser compressor stage was adjusted to obtain the same minimum pulse duration of the pump beam on the copper sample.

**Betatron radiation.** The laser pulse focused in the gas jet creates an underdense plasma. Electrons from this plasma are accelerated in the wakefield of the laser, reaching energies of a hundred MeV. They follow an oscillating trajectory with typical transverse amplitude of 1 μm and longitudinal period of 150 μm. Due to this relativistic motion, they emit in the forward direction a low-divergence X-ray beam (~10 mrad) with a continuous spectrum extending up to about 10 keV[11]. This is called the betatron radiation. We made use of the ionization injection scheme[26], ensuring the production of stable betatron radiation[11]. Typically, $10^5$ photons/shot/ 0.1%BW are emitted around 1 keV[27].

**X-ray pulse duration.** Since the accelerated electron bunch is confined just behind the laser pulse, the betatron source intrinsically provides ultrashort pulses. We calculated the temporal shape of the betatron X-ray pulse using a particle-in-cell simulation[28]. For our parameter regime, accelerated electrons originate from the peak of the laser field only, where the L-shell of the nitrogen atoms is ionized: this is the so-called ionization injection. This localization ensures the production of a single electron bunch in the first wakefield cavity (see inset of Fig. 5). The simulation yields a 6-fs-long electron bunch (FWHM). The computed electron trajectories allow to then calculate the X-ray temporal profile at 930 eV, corresponding to the energy of the copper $L_3$ absorption edge. The result is shown in Fig. 5, and gives an FWHM duration of 9 fs. This is slightly longer than the electron bunch duration, due to the "slippage" of the emitted photons traveling faster than the wiggling electrons. This effect is even emphasized for lower energy electrons, which need larger oscillations for emitting the same photon energy. These electrons, injected later in the accelerating cavity and thus positioned at the back end of the bunch, are responsible for the tail of the X-ray pulse profile that can be seen in Fig. 5.

**X-ray spectrometer.** A Bragg spectrometer was built for measuring the absorption spectra. It covers a 50 eV bandwidth centered around the L-edges of Cu ($L_3$ at 932 eV and $L_2$ at 952 eV). It consists of a toroidal RbAP crystal and a CCD camera

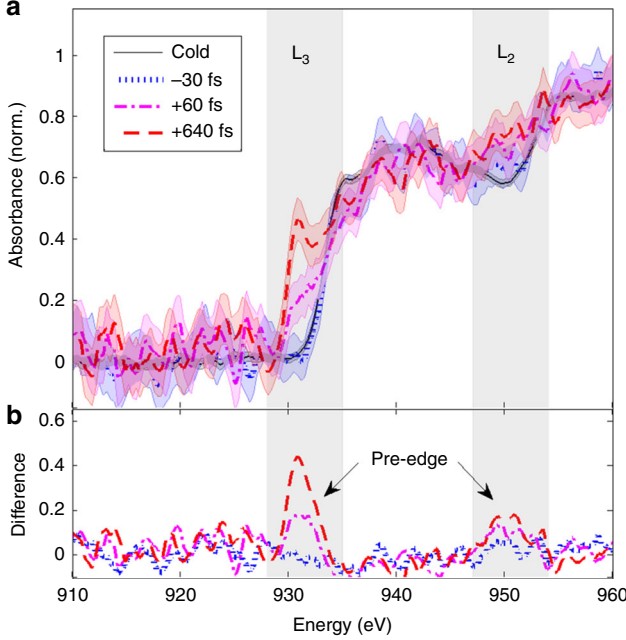

**Fig. 3** Time-resolved XAS data. Selection of some measurements near the Cu L-edges ($L_3$ and $L_2$), without and with the pump pulse (incident fluence of 1 J cm$^{-2}$), for three different X-ray probe delays. **a** Normalized X-ray absorbance. The shadowed area indicates the standard deviation of the measurements over a series of 50 consecutive shots. **b** Differential absorbance with respect to the curve without pump. Clear pre-edges appear a few eV below the L-edges just after the sample heating

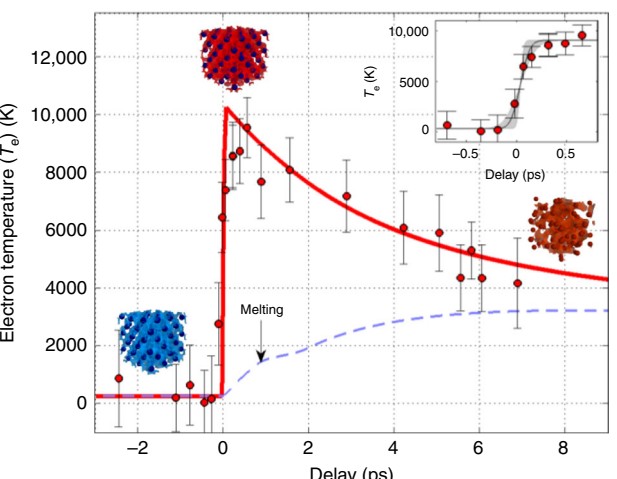

**Fig. 4** Time evolution of the electron temperature. The data deduced from time-resolved XAS measurements (full circles) are compared with the two-temperature hydrodynamic calculation (plain line). The calculated ion temperature is also plotted (dashed line). The incident fluence is 1 J cm$^{-2}$. The experimental data indicate a rise-time of $75 \pm 25$ fs (line and shadowed area in the inset). The gradual decrease observed at longer time is well reproduced by the model and is understood as the electron−ion thermal equilibrium. Error bars are calculated from the integrated standard deviation within the pre-edge in the XAS data series

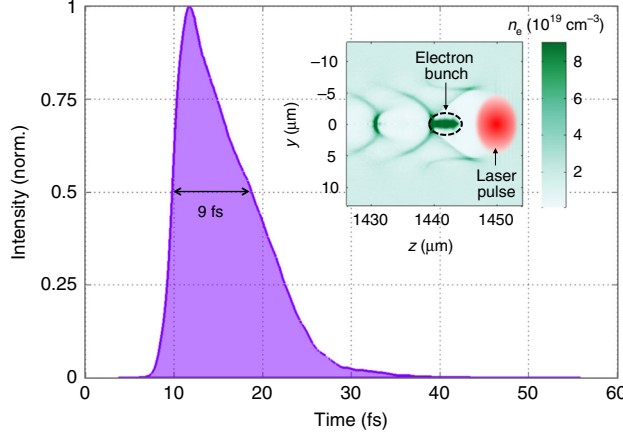

**Fig. 5** Result of particle-in-cell simulations. Main curve: temporal X-ray profile calculated on-axis for 930-eV photons. Inset: Two-dimensional map of the plasma density $n_e$, showing the electron bunch accelerated in the wake of the laser pulse

placed at its sagittal focus, which coincides with the Rowland circle in order to have a spectral resolution independent from the size of the X-ray source. A slit placed at the tangential focus of the crystal allows for noise removal due to the fluorescence of the crystal. The CCD is an in-vacuum water-cooled PI-MTE from Princeton Instruments®. The crystal is curved with a sagittal radius of 85 mm, a tangential radius of 200 mm and was built by Saint-Gobain Crystals®.

**Sample and procedure.** A $70 \pm 10$ nm Cu layer was deposited on a $30 \times 80$ mm$^2$ area by evaporation on a 1-µm Mylar™ foil strengthening the whole membrane. Since the copper heated area is ablated after a single laser irradiation, the sample is moved to present a fresh surface shot after shot, allowing ~500 shots per membrane. An automatized procedure was implemented to trigger the laser shot, command the X-ray spectrometer acquisition and move the sample, with an effective repetition rate of 0.3 Hz. For sufficient data statistics and signal to noise ratio, 50 acquisitions were required per spectrum, completed by series of raw spectra (without sample) for normalization. After a pump-probe series, the ablated areas were scanned a second time by the X-rays in order to check the correct overlap between the pump and the probe beams.

**XAS data extraction.** In order to recover each absorption spectrum, we registered three spectra (each one resulting from a series of 50 shots): without sample (reference), with cold sample, and with heated sample. For each series, a median filter is applied in order to remove the residual hot spots noise. The 50 images are summed and the spectra are extracted from a line-out profile. The cold/heated transmissions are obtained from the division of the cold/pumped series over the reference one. The cold absorption spectrum (coming from the logarithm of the transmission) is then set to zero below the $L_3$ edge, and normalized above the $L_2$ edge. The same normalization is used for the heated spectrum. The remaining error bars are mainly limited by the number of detected X-ray photons. Several spectra were measured under similar conditions (incident heating fluence and delay) in order to increase the statistics and reduce the error bars.

**Molecular dynamics simulations.** Molecular dynamics simulations were carried out with the ab initio plane wave density functional theory (DFT) code Abinit[29]. DFT is applied together with local density approximation[30]. Simulations are performed in the framework of the projected augmented wave method[31,32]. All calculations are made with a $3 \times 3 \times 3$ Monkhorst-Pack k-point mesh. The simulations compute the copper absorption spectrum in the XANES region, reproducing the pre-edge structure when electrons are hot. A recent study showed the univocal correlation between the pre-edge amplitude and the electron temperature[23].

**TTM simulations.** The two-temperature model (TTM) is used to calculate the evolution of the electron and ion temperatures $T_e$ and $T_i$[33]. It is integrated in the one-dimensional hydrodynamic code ESTHER detailed in ref.[34], that describes the matter evolution with multiphase equation of state[35] and consistent ion heat capacity. The electron heat capacity and electron−ion coupling parameter are taken from Lin et al.[36]. A simplified approach is here considered for the laser energy absorption (homogeneous heating). The source term is a 30-fs FWHM Gaussian function with 0.15 J cm$^{-2}$ integrated fluence, that corresponds to 15% overall laser absorption calculated by solving the Helmholtz equations with an incident fluence of 1 J cm$^{-2}$. The time required to homogenize the electron temperature along the sample thickness is not considered in this simulation. It is retrieved from the best fit of the experimental $T_e$ rise-time $\tau_{rise}$ by a convolution of the TTM calculation with a Gaussian function.

**Data availability.** The data that support the findings of this study are available from the corresponding author upon reasonable request.

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

## Acknowledgements

We thank Martine Millerioux for sample preparation and Vanina Recoules for support with the Abinit code. This work was funded by the Agence Nationale pour la Recherche through the FEMTOMAT Project No. ANR-13-BS04-0002 and the European Research Council through the X-Five grant (Contract No. 339128).

## Author contributions

B.M. and K.T.P. conceived the betatron source. L.L. designed the X-ray spectrometer. J.-P.G. operated the laser system. F.D., N.J., L.L., B.M. and K.T.P. carried out the

experiment and analyzed the data. N.J. and A.L. performed simulations. F.D., L.L., P.R. and K.T.P. led the project. B.M. wrote the manuscript and all authors contributed to its improvement.

## Additional information

**Competing interests:** The authors declare no competing interests.

