## [Peer Review File · Nature Communications]

Editorial Note: This manuscript has been previously reviewed at another journal that is not operating a transparent peer review scheme. This document only contains reviewer comments and rebuttal letters for versions considered at Nature Communications. Parts of this peer review file have been redacted as indicated to remove third-party material where no permission to publish could be obtained.

Reviewers' comments:

Reviewer #1 (Remarks to the Author):

Mahieu et al. present the X-ray Absorption Spectrum (XAS) of warm dense copper with femtosecond resolution. The authors have accomplished a first, an XAS of warm dense matter with this time resolution. In the main result, the rise time of the electron temperature is explained by including the ballistic transport of the electrons. This mechanism of energy diffusion has been commonly employed in the warm dense matter community. In this reviewer's view, this manuscript meets the Nature Communications criteria that "the results are novel, and the manuscript is important to scientists in the specific field."

One more detailed comment follows.

In the discussion of alternative sources, it is stated that "the number of shots needed to recover a single absorption spectrum limits the range of systems and regimes that can be studied (continuously renewed targets, e.g. in liquid jets, or reversible phase transitions)." It is implied that X-ray FEL experiments are limited to continuously renewed targets. However, the reference 10 (Gaudin et al.) demonstrated that X-ray absorption studies of non-reversible processes can be studied with X-ray FELs.

Reviewer #3 (Remarks to the Author):

The authors have revised the manuscript following the recommendations from the previous referees, however, there still remains several unanswered questions which prevents me recommending this for publication. I summarise here my main critique:

1. Despite been asked by one of the referees, the authors continue to omit any description of their DFT simulations. These are needed in order to understand Figure 2, and also DFT simulations are required to convert the position of the pre-edge into a temperature measurement. They cite Ref. 21 - but in 21 a different code (expresso) was used, not Abinit as in the current work. Without this information is basically impossible to assess anything about the work presented here.

2. I am rather sceptical of the discussion presented on the top of page 8 regarding the electron velocity. If I take the pump laser parameters, the Cu foil is illuminated with a laser of intensity mid 10^{14} W/cm². At these intensities, hot electrons with energies >1 keV are produced (see for example the old review from C Max - <https://www.osti.gov/biblio/5630914>). These electrons are the main carrier of the heat flux, not the bulk electrons. The two-temperature hydrodynamic model does not capture this process (as far as discussed in the manuscript), so I am not convinced the interpretation they offer is sound.

In summary, this manuscript still requires substantial changes before it can be accepted in Nature Communications.

Reviewers' comments

Reviewer #1

Mahieu et al. present the X-ray Absorption Spectrum (XAS) of warm dense copper with femtosecond resolution. The authors have accomplished a first, an XAS of warm dense matter with this time resolution. In the main result, the rise time of the electron temperature is explained by including the ballistic transport of the electrons. This mechanism of energy diffusion has been commonly employed in the warm dense matter community. In this reviewer's view, this manuscript meets the Nature Communications criteria that "the results are novel, and the manuscript is important to scientists in the specific field."

One more detailed comment follows.

In the discussion of alternative sources, it is stated that "the number of shots needed to recover a single absorption spectrum limits the range of systems and regimes that can be studied (continuously renewed targets, e.g. in liquid jets, or reversible phase transitions)." It is implied that X-ray FEL experiments are limited to continuously renewed targets. However, the reference 10 (Gaudin et al.) demonstrated that X-ray absorption studies of non-reversible processes can be studied with X-ray FELs.

We are glad to see that Reviewer #1 is rather positive.

Ref. 10 indeed demonstrated that X-ray absorption studies of non-reversible processes can be studied with X-ray FELs. We agree on this point, but the writing of the text could indeed lead to misunderstanding. We thus modified the paragraph accordingly. In particular, one can now read the sentence : « *On XFEL, investigation of a non-reversible process was albeit reported by means of dispersive measurements, who remain in any case limited to a few eV bandwidth and must circumvent the inherent shot-to-shot spectral fluctuations of such a source [...]* ».

We thank Reviewer #1 for this comment.

Reviewer #3

The authors have revised the manuscript following the recommendations from the previous referees, however, there still remains several unanswered questions which prevents me recommending this for publication. I summarise here my main critique:

1. Despite been asked by one of the referees, the authors continue to omit any description of their DFT simulations. These are needed in order to understand Figure 2, and also DFT simulations are required to convert the position of the pre-edge into a temperature measurement. They cite Ref. 21 - but in 21 a different code (expresso) was used, not Abinit as in the current work. Without this information is basically impossible to assess anything about the work presented here.

We agree with Reviewer #3 that a description of DFT simulations is necessary for the clarity of the manuscript. We apologize for this omission in the previous version.

We thus propose to add a dedicated paragraph in the Methods section:

« *Molecular dynamics simulations*

Ab-initio molecular dynamics simulations were carried out with the ab-initio plane wave density functional theory (DFT) code Abinit29. DFT is applied together with local density approximation [30]. Simulations are performed in the framework of the projected augmented wave method [31,32]. All calculations are made with a 3x3x3 Monkhorst-Pack k-point mesh.

The simulations compute the copper absorption spectrum in the XANES region, reproducing the pre-edge structure when electrons are hot. A recent study showed the univocal correlation between the pre-edge amplitude and the electron temperature [23].»

with new references:

- [30] Perdew, J. P. & Wang, Y. Accurate and simple analytic representation of the electron-gas correlation energy. *Phys. Rev. B*, 45, 13244 (1992).
- [31] Blöchl, P. E. Generalized separable potentials for electronic-structure calculations. *Phys. Rev. B*, 41, 5414 (1990).
- [32] Torrent, M., Jollet, F., Bottin, F., Zérah, G. & Gonze, X. Implementation of the projector augmented-wave method in the ABINIT code: Application to the study of iron under pressure. *Comput. Mater. Sci.*, 42, 337–351 (2008).

The measured pre-edge integral, which we obtain experimentally, is converted into electron temperature via correlation with DFT simulations. This approach was formerly proposed and validated by Cho et al., who made use of the quantum-espresso package [Paolo, G. *et al.* QUANTUM ESPRESSO: a modular and open-source software project for quantum simulations of materials. *J. Phys. Condense. Matter* **21**, 395502 (2009)]. We use in our case the Abinit code [17] for DFT simulations.

A systematic study of the link between calculated electron temperature and pre-edge integral is presented in our recent publication [Jourdain et al., Electron-ion thermal equilibration dynamics in femtosecond heated warm dense copper, in *Phys. Rev. B* (2018), added to references of the current manuscript]. A quasi-linear regression law was found, as shown in the additional Figure below. Equilibrium, out-of-equilibrium and liquid phases were investigated, assessing the validity and robustness of the method.

The writing of a complementary article, dedicated to the complete description of XANES computations in the specific case of warm dense copper, is also in progress.

[Redacted]

[Redaction]

2. I am rather sceptical of the discussion presented on the top of page 8 regarding the electron velocity. If I take the pump laser parameters, the Cu foil is illuminated with a laser of intensity mid 10^{14} W/cm^2 . At these intensities, hot electrons with energies $>1 \text{ keV}$ are produced (see for example the old review from C Max - <https://www.osti.gov/biblio/5630914>). These electrons are the main carrier of the heat flux, not the bulk electrons. The two-temperature hydrodynamic model does not capture this process (as far as discussed in the manuscript), so I am not convinced the interpretation they offer is sound.

The former review of C. Max applies to a coronal plasma of several hundreds of eV in which the laser absorption is dominated by the inverse bremsstrahlung and where nonlinear phenomena can lead to the creation of electrons with energies of several keV. This is a regime different from the one presented in our study.

In the experiment, the pump laser fluence was $F \sim 1 \text{ J/cm}^2$ (the maximum attainable value was 6.5 J/cm^2 in our configuration). With a duration of 30 fs, it produces an intensity on target of $3 \cdot 10^{13} \text{ W/cm}^2$. Since the energy absorption calculated by Helmholtz equations is $\sim 15\%$, the effective intensity boils down to $I = 0.5 \cdot 10^{13} \text{ W/cm}^2$. Considering a free electron in this laser field, its average kinetic energy is given by the ponderomotive energy $U_p = \frac{e^2 A^2}{4m\omega^2} = \frac{e^2 I}{2c\epsilon_0 m\omega^2} = 0.3 \text{ eV}$ (e is the electron charge, A the laser amplitude, m the electron mass, ω the laser central frequency, c the light velocity and ϵ_0 the vacuum permittivity).

In our experimental conditions, we thus don't expect to produce electrons with energies above the eV range, which could indeed modify the heat flux process.

Additionally, to evaluate the average electron temperature, one can consider the conservation of energy from the incident pump laser to the excited electrons. The volume of excitation is $V = S \cdot d$, with $d = 70 \text{ nm}$ the sample thickness and $S \sim \pi \cdot 400 \mu\text{m}^2$ the area of excitation. The number of electrons within this volume is $N = \rho \cdot V$, $\rho = 8.49 \cdot 10^{28} \text{ m}^{-3}$ being the electron density in solid copper. The deposited energy is $E = F \cdot S$, and the effective absorbed energy is $E_{eff} = 0.15 \cdot E$ (15% of absorbed energy calculated by Helmholtz equations).

According to the equipartition of energy, the statistical electron temperature is given by $T = \frac{1}{N} \cdot \frac{2}{3} \cdot \frac{E_{eff}}{k_B} = \frac{2}{3} \cdot \frac{0.15 \cdot F}{\rho \cdot d \cdot k_B} = 1.05 \text{ eV}$ (where k_B is the Boltzmann constant), i.e. in line with the values of electron temperature presented in our manuscript.

In summary, this manuscript still requires substantial changes before it can be accepted in Nature Communications.

We hope our answers and changes to the text will convince Reviewer #3.

REVIEWERS' COMMENTS:

Reviewer #3 (Remarks to the Author):

I am happy with the changes to the manuscript which now properly addresses all of my previous criticisms. I don't have any further comments and I recommend publication.

Reviewers' comments

Reviewer #3: « I am happy with the changes to the manuscript which now properly addresses all of my previous criticisms. I don't have any further comments and I recommend publication. »

The single Reviewer's comment is definitely positive and we are pleased to note that our previous reply convinced the Referees so that final recommendation is positive.